# Biofilm and Equine Limb Wounds

**DOI:** 10.3390/ani11102825

**Published:** 2021-09-27

**Authors:** Elin Jørgensen, Thomas Bjarnsholt, Stine Jacobsen

**Affiliations:** 1Costerton Biofilm Center, Department of Immunology and Microbiology, Faculty of Health and Medical Sciences, University of Copenhagen, DK-2200 Copenhagen, Denmark; tbjarnsholt@sund.ku.dk; 2Department of Clinical Microbiology, Rigshospitalet, DK-2100 Copenhagen, Denmark; 3Department of Veterinary Clinical Sciences, Faculty of Health and Medical Sciences, University of Copenhagen, DK-2630 Taastrup, Denmark; stj@sund.ku.dk

**Keywords:** biofilm, wound, infection, horse, bacterial aggregates, wound chronicity, delayed wound healing

## Abstract

**Simple Summary:**

Delayed wound healing commonly occurs in limb wounds of horses. These equine limb wounds share many similarities with chronic wounds in humans and one of them seems to be the presence of biofilm, even though equine wound biofilm research is just emerging. Biofilms are aggregates of bacteria, and within these aggregates, the bacteria are protected from both antimicrobial substances and the immune response of the host. Biofilm infections in wounds often delay healing and are impossible to detect with routine diagnostics. However, if suspected, aggressive treatment is needed and includes physically removing biofilm and unhealthy tissue from the wound during debridement and immediately applying antimicrobial compounds to kill any biofilm or bacteria not removed during debridement.

**Abstract:**

In chronic wounds in humans, biofilm formation and wound chronicity are linked, as biofilms contribute to chronic inflammation and delayed healing. Biofilms are aggregates of bacteria, and living as biofilms is the default mode of bacterial life; within these aggregates, the bacteria are protected from both antimicrobial substances and the immune response of the host. In horses, delayed healing is more commonly seen in limb wounds than body wounds. Chronic inflammation and hypoxia are the main characteristics of delayed wound healing in equine limbs, and biofilms might also contribute to this healing pattern in horses. However, biofilm formation in equine wounds has been studied to a very limited degree. Biofilms have been detected in equine traumatic wounds, and recent experimental models have shown that biofilms protract the healing of equine limb wounds. Detection of biofilms within wounds necessitates advanced techniques that are not available in routine diagnostic yet. However, infections with biofilm should be suspected in equine limb wounds not healing as expected, as they are in human wounds. Treatment should be based on repeated debridement and application of topical antimicrobial therapy.

## 1. Introduction

Just like humans, horses also suffer from chronic limb wounds. In both species, hypoxia and chronic inflammation have been suggested as main contributors to impaired healing [1,2,3,4,5,6]. Within the last decade, infections with biofilm have been investigated in chronic wounds in humans [7,8] and are now considered ubiquitous within these wounds [9,10]. A possible link has been established between wound chronicity and biofilm presence [7,8,9,10]; however, the exact mechanisms by which biofilms impair wound healing have not yet been fully elucidated.

In veterinary medicine, infections with biofilm in relation to wounds have been studied to a very limited degree. The presence of biofilms has been reported in a number of wounds in three equine studies [11,12,13] and in three canine studies [14,15,16]. We have recently shown in an equine experimental wound model that biofilms were present in equine limb wounds but not in body wounds [17]. Further, in a recent equine experimental wound model with bacterial inoculation, we found biofilms to negatively affect wound healing of distal limb wounds but not body wounds [18]. Based on these initial findings of biofilms in equine limb wounds and on several similarities between equine and human impaired wound healing [19,20,21], there is a good reason to expect biofilm formation to also play a role in the delayed healing of limb wounds in horses. Additional basic knowledge is needed to investigate prevalence and aetiology including how biofilm infections occur and contribute to delayed equine limb wound healing. The purpose of this review is to increase the awareness of biofilms in relation to wounds and their potential consequences for wound healing among veterinary, especially equine, clinicians and practitioners.

## 2. Biofilm in General

Bacteria can live either as single cells, also known as planktonic cells, or as biofilm (Figure 1). Most of our knowledge of bacteria originates from classic culture plates and shaken cultures where the bacteria live as planktonic cells. Nevertheless, biofilm formation is an ancient aspect of bacterial life and one of the reasons bacteria can be found throughout nature, even in extremely hostile and depleted environments. Thus, biofilm is the default mode of bacterial life. Unfortunately, biofilm is a more complicated lifestyle to study than planktonic cells [22] and the majority of our knowledge comes from in vitro studies of biofilms. However, biofilms in vivo are markedly different from in vitro biofilms in e.g., size, organisation, and transcription profile [23,24,25]. A recent international consensus statement [26] defined biofilm as:
“A structured community of microbes with genetic diversity and variable gene expression (phenotype) that creates behaviours and defences used to produce unique infections (chronic infection). Biofilms are characterised by significant tolerance to antibiotics and biocides while remaining protected from host immunity”.

The matrix of biofilms has mainly been investigated in vitro and has been found to consist of extracellular polymeric substances, which are mainly polysaccharides, proteins, DNA, and lipids. The matrix provides the bacteria within the biofilm with protection from environmental changes and gives mechanical stability. Host components, like released neutrophil DNA and host extracellular matrix material, can be embedded in the matrix in in vivo situations [27,28]. Furthermore, the matrix is a nutrient source and a substrate for enzymatic activity, hydration, and horizontal gene transfer between the embedded bacteria [29]. Bacteria living together in biofilms in vitro can deploy a cell-cell communication system, named quorum sensing (QS), to coordinate the behaviour of the individual cells and thereby gain an array of advantages compared with planktonic bacterial cells [30]. The QS system is activated when the cell density increases, the quorum size is reached, and the system will regulate the expression of virulence factors, antimicrobial resistance, tolerance, and biofilm formation and maintenance [31,32]. Whether, when, and how the QS system is employed in in vivo infections is unknown. One study showed lower QS expression during in vivo infections compared to laboratory conditions [23], underlining the marked difference between in vitro and in vivo biofilms, and the need for more in vivo biofilm research. Further, as potential new anti-biofilm treatments, which target the quorum sensing [31,33], are being investigated, it is even more important to know how and to which extent in vivo biofilms use quorum sensing.

This “stronger-together” mode of living makes biofilms markedly less susceptible to antimicrobial drugs through development of tolerance. Tolerance is a transiently decreased susceptibility to antimicrobials achieved with the biofilm phenotype, and is not to be confused with antimicrobial resistance, a genetically inherited trait. Tolerance takes time to achieve, and as a result, immature biofilms (less than 1–2 days) are easier to treat successfully with antimicrobials than mature biofilms; this has been described as a time dependent therapeutic window [34,35]. Mature in vitro biofilms can tolerate up to 1000-fold higher antibiotic concentrations than planktonic cells [36,37].

Biofilms are “experts” in “distracting” the immune system of the host by supressing a proper and effective inflammatory response in several ways [38,39]. During insufficient inflammation, host cells, especially neutrophils, release large quantities of enzymes and oxidative radicals that cause collateral damage and exert deleterious effects on the surrounding tissue [40]. Further collateral damage occurs as antibodies incapable of binding to the bacteria form immune complexes in the tissue and activate both opsonisation and the complement system [41]. In addition, biofilm infections in chronic wounds cause low oxygen tension, as the biofilm itself consumes oxygen, as do the attracted leukocytes [42]. This leads to a cycle of hypoxia and chronic inflammation favouring further biofilm formation and collateral damage, thereby maintaining the chronicity of the wounds.

Nonetheless, biofilms can cause chronic infections, but they are not always bad. Bacteria organised as biofilms are critically important for, e.g., the gastrointestinal tract and skin health of both humans and animals. In these settings, the bacteria boost immune defences, help defend the host, and even participate in the digestion of nutrients otherwise not degradable by the host [43,44].

In human medicine, biofilms contribute to a wide variety of tissue infections apart from wound infections. Urinary tract infections, osteomyelitis, device-related infection, chronic otitis media, chronic sinusitis, endocarditis, dental plaques, kidney stones, and lung infection (especially in cystic fibrosis patients) are just part of the repertoire of biofilms [45,46]. To date, very little is known about how biofilms are established in clinical infections, nor about when or how they become difficult to eradicate. In the US, infections with biofilm are estimated to account for 17 million infections and 550,000 deaths every year, costing the health system close to 100 billion USD [47]. Biofilms have been suggested to be one of the greatest challenges of the modern medical world [48] and intensive research into how to improve the diagnostics, prevent biofilm formation, and eradicate infectious biofilms is ongoing.

There is only a very sparse bulk of literature on biofilm-related infections and their diagnosis and management in veterinary clinical medicine [11,12,13,14,15,16,17,18,49,50,51]. To understand the subject, we currently have to broaden our view and look to other scientific areas (such as experimental medicine, biofilm research, and human clinical medicine). As many animal models are used in biofilm research, including rodents, rabbits, pigs, horses, etc., it seems plausible (in lieu of hard and fast scientific evidence) that biofilms could also be involved in many chronic infections in veterinary patients similar to what has been described in man.

## 3. Similarities between Equine and Human Wounds with Delayed Healing and the Potential to Use the Horse as a Model

Equine limb wounds are among the most frequent injuries in horses [52,53,54,55], and these wounds often need to heal by second intention due to tissue loss, contamination, and/or failure of primary closure [56,57]. Biofilm is a contributing factor to delayed healing in human chronic wounds like diabetic foot ulcers and venous leg ulcers. Infections with biofilm might likely also be a contributing factor for delayed limb wound healing in horses, as equine and human chronic wounds share many similarities [19,20,21,57,58,59,60]:Location: Tendency to affect lower extremities: delayed healing mainly occurs on the limb or foot in both humans and horses [3,17,20,61,62,63];Healing type: wounds mainly heal by epithelialisation, while contraction only accounts for approximately 15–20% in equids [6,64,65] and approximately 30% in humans [66];Inflammation: both equine limb wounds and chronic wounds in humans are often arrested in a weak and prolonged inflammatory state with dysregulated cytokine profiles [57,67,68,69,70,71];Oxygen supply: decreased blood flow, hypoxia, and ischaemia occur in wounds with delayed healing in both humans and horses [4,72,73];Increased protease activity: increased protease activity occurs in wounds with delayed healing in both humans and horses [19,71,74];Wound pathogens: *Staphylococcus aureus* and *Pseudomonas aeruginosa* are common wound pathogens in both humans and horses [11,13,75,76,77].

The protracted healing of chronic wounds in man and horses is difficult to mimic, as many animals models have markedly different healing patterns from the delayed healing pattern seen in man and horses. Popular animal models for human wounds are rodents and rabbits with thin and loose skin that heal mainly by contraction caused by their cutaneous trunci muscle [78,79]. Some of the biggest challenges in the creation of representative animal models are to mimic the low-grade inflammation and the epithelialisation-dominated healing pattern that occur in chronic wounds [80]. Another more relevant, however less used, model animal in wound research is the pig, as porcine wound healing resembles human wound healing to a larger extent than rodents and rabbits because pigs are more tight-skinned and heal predominantly by epithelialisation [81,82,83]; accordingly porcine wound models are considered superior to other animal models due to their translational value [59,83,84]. However, disadvantages of the porcine models include that the animals are relatively young, a lack of wound exudate, lack of true chronicity, and the fact that wounds are situated on the flanks or backs of the pigs, regions that do not display the peripheral circulation disorders most often present in chronic foot and limb ulcers in humans [83]. In vivo models are necessary when investigating biofilms’ effect on healing and effect of potential anti-biofilm treatments. However, it is unclear to what extent these results can be translated to the clinical setting due to host responses, different wound pathogenesis, and a lack of “true” wound chronicity in the animal models [3,85]. Furthermore, most animal models are young, growing animals without comorbidities that do not resemble the human patient population, as chronic wound patients are normally elderly and often with comorbidities such as diabetes, cardiovascular disease, and/or obesity. The horse has been suggested to be a more relevant model animal species, as healing of limb wounds in horses possesses some of the same inherent issues, e.g., slow and complication prone healing, as wounds on the extremities of humans [57]. Similar to findings in chronic wounds in humans [9,10], biofilms are widespread in equine experimental limb wounds [17,18], and their presence delays healing in limb wounds. Further, as equine body wounds are not affected by biofilm presence, body wounds in the same animal can serve as an internal control [18]. Potential investigations using the horse as a model animal for human chronic wounds would also benefit the horse and its own healing issues in relation to infections with biofilm.

## 4. Biofilm and Equine Limb Wounds

During the last decade, biofilms have been reported in equine surgical and traumatic wounds in three studies [11,12,13], however as suboptimal detection methods were used in these studies (i.e., histology and gram stain, see section below on detection methods), it is difficult to know the true prevalence of biofilm in equine traumatic limb wounds. Nevertheless, in these studies the prevalence of biofilm in equine traumatic wounds was reported to be 10% [12], 61.5% [13], and “the majority of the wounds” (n = 8) [11], respectively. No investigations into the effect of biofilms on wound healing were carried out in these studies of traumatic patient wounds. In 2010, Westgate and colleagues speculated that it is unknown how the bioburden and potential presence of bacterial biofilm affect equine wound healing, though a negative influence would be expected [86].

In an experimental equine study, our group found that biofilms were present in 100% of “clean”(surgically prepared and bandaged) equine limb wounds, but not in body wounds; gold standard detection methods (peptide nucleic acid (PNA) fluorescence in situ hybridization (FISH) and confocal laser scanning microscopy (CLSM), as described below) were used in this study [17]. Thus, this study indicated that biofilms might be involved in the delayed healing of limb wounds. Equine limb wounds’ proximity to the ground might increase the risk of bacterial contamination in traumatic wounds, this might however not necessarily have a negative influence on wound healing, as an equine experimental wound study showed that faeces contamination improved limb wound healing [87]. Reasons for the differences in biofilm presence in our experimental equine body versus limb wounds could be the diminished and prolonged inflammatory response detected in limb wounds compared with body wounds [1,18,67]. Another reason could be the hypoxic conditions of equine limb wounds [2,4], which reduces resistance to infections [88]. Impaired inflammation and hypoxia might therefore result in favourable conditions for bacteria to settle as biofilms. However, both conditions are also effects of biofilm infections [42,89], thus highlighting a classic “chicken-or-egg” causality dilemma.

Our group further developed an equine experimental wound model involving inoculation with *S. aureus* and *P. aeruginosa*, to study the potential biofilm formation and its effect on healing. Biofilms that formed in inoculated limb wounds persisted and negatively affected healing, while biofilms that formed in inoculated body wounds were quickly eradicated and did not affect healing [18]. This shows that it is not necessarily the presence of bacteria in a wound that is the problem, whether an infection with biofilm occurs is determined by the receiving wound bed and its ability to fight off bacterial biofilms [10,84], which are markedly different for limb and body wounds in horses.

The biofilm forming potential of isolated bacteria has been tested in vitro for equine wound isolates [13,90]. However, there is no correlation between a bacteria’s biofilm forming potential in vitro and the actual presence of biofilms in the wound the bacteria were cultured from [23,25], so to better understand biofilms and the effect on equine wound healing other tests and diagnostics must be used, as discussed below.

## 5. Handling Biofilms in Wounds

As research on biofilms in equine wounds is just starting to emerge, the following subsections are mainly based on the knowledge from how biofilms are handled in human chronic wounds. However, due to the aforementioned similarities, we find this information highly relevant and applicable for equine wounds as well.

### 5.1. Detection of Biofilm in Wounds

Diagnosing biofilms in wounds is complicated by (1) a lack of evident clinical signs, (2) heterogeneous distribution of biofilms and bacteria in wounds, (3) viable, but non-culturable, bacteria, and (4) a requirement for advanced microscopy methods.

Biofilms in equine and human wounds are small (5–200 µm) and not visible to the naked eye [7,8,17,24,77]. Infection with biofilm is not necessarily accompanied by the cardinal signs of inflammation such as pain, redness, heat, and swelling, and many studies have tried to identify specific clinical signs related to biofilm-infected chronic wounds. However, signs vary, and no specific clinical signs have been found to correlate with biofilm infection [10,18,26,91,92,93], which makes it extremely difficult for clinicians to evaluate whether wounds are infected with biofilm.

A main problem when investigating bacterial burden in wounds, is that biofilms as well as single bacterial cells are heterogeneously distributed in the wounds [10,24,94], and thereby samples from different parts of a wound can reveal different results [77,94,95]. As a result, there is a risk of obtaining misleading, inconclusive, or even negative results when sampling from a chronic wound. Sampling from more than one site is generally recommended if the wound size allows [9,95].

It is challenging to diagnose wound infections with biofilms using culture methods. Standard clinical culture techniques do not provide information on whether the isolated bacteria were present as planktonic cells or biofilms in the wound [96,97]. Furthermore, biofilm bacteria will often be underrepresented or potentially not detected at all on routine culture, as bacteria in biofilms can have a markedly reduced growth rate [77,94,98,99].

Biopsies are recommended as sample material when evaluating bacterial burden in wounds [97,100], as swabs pose a risk of including irrelevant surface contamination. If obtaining a biopsy is not an option, a swab should be obtained carefully after debridement. The Levine technique is considered the superior method for obtaining wound swabs [12,101,102] and can easily be applied to equine wounds. With this technique, the swab is rotated over a 1 cm^2^ area of the wound applying sufficient pressure to release wound fluid (and thereby bacteria) from deeper parts of the wound [103]. As mentioned above, several samples are recommended for larger wounds due to the heterogenetic distribution of bacteria and biofilm in wounds. For both sample methods (biopsy and swab), it is relevant to do a (semi-)quantitative culture with identification of all bacteria to get an idea of the burden/quantity of the different bacteria in the wound. Before sampling, it is recommended to cleanse or debride the surface to remove surface contamination, as the bacteria at the wound surface are rarely responsible for the wound infection [102,104]. Avoiding cotton-tipped swabs is also recommended, as fatty acids in cotton swabs can potentially inhibit growth for some fastidious bacteria [102]. A useful swab, also in equine wounds, is the Copan ESwab that has great bacterial uptake and recovery [105].

When determining the bacterial burden in wounds by culture, sonication and/or repeatedly vortexing of the sample is recommended in order to disrupt any potential biofilms [10].

Using molecular techniques (e.g., qPCR and 16S ribosomal ribonucleic acid (rRNA) gene sequencing) will ensure that even slow-growing bacteria are detected, but these methods do not provide information on the bacterial phenotype (i.e., planktonic versus biofilm) [96,97]. Shotgun sequencing is a deeper sequencing technique that can give more information on strain level and detect so-called genetic signatures of biofilm formation [106], however, this is not a gold standard method for detection of biofilms. Anyways, molecular techniques will give more relevant information about the bacterial composition of the wound [107], by indicating the predominant species towards which therapy should be directed [10].

To detect biofilms in wounds, at least for now, it is necessary to directly visualise the aggregates of bacteria in the wound tissue and the immune reaction towards the biofilms (Figure 1). CLSM and scanning electron microscopy (SEM) are considered gold standard techniques for biofilm detection in wounds [97,108]. Peptide nucleic acid (PNA) fluorescence in situ hybridization (FISH) combined with CLSM is a very useful method for detecting bacteria in wound samples, as PNA probes bind specifically to bacterial 16S rRNA and the fluorescence can then be detected by CLSM [94,109]. PNA FISH and CLSM have been successfully used to detect biofilm in equine wound samples [17,18].

PNA FISH followed by CLSM and SEM are advanced and time-consuming methods used in research, and they are not routinely available in diagnostic laboratories for either human or veterinary patient samples. Currently in human medicine, biofilm infections are therefore a clinical diagnosis, where biofilms should be suspected in wounds that do not heal in a timely manner despite appropriate treatment, and therefore anti-biofilm treatment should be initiated accordingly [9,10]. A similar approach is recommended for equine wounds [110]. Developing tools to diagnose biofilms in a clinical setting is naturally an important research topic [111], and many ideas are being investigated including e.g., blotting/staining methods [112] and detection of volatile organic compounds released by bacteria in biofilms [113]. Ideally, a future diagnostic tool would be able to detect the bacterial species present and whether or not these are present as biofilms [10,93].

### 5.2. Treatment of Biofilm in Wounds

Per definition, biofilms exhibit tolerance toward antimicrobials and withstand host immune responses [108], making successful treatment of biofilm infections in wounds challenging. Systemic antibiotic treatment has no proven effect on human chronic wound infections, but can be used to prevent systemic spread or acute local infection with clinical signs of inflammation in the tissues around the wound [85,114,115,116]. Similarly, systemic antibiotics have no effect on the bacterial burden in granulating wounds [117,118], and should therefore mainly be used for acute wound infections. Preventing formation of biofilm should be a priority. Administering antimicrobials (locally and/or systemically) to horses with wounds must thus commence as early as possible, before bacteria have gone from their planktonic to their biofilm state. This has been shown to occur within approximately 24 h [119].

The cornerstone of biofilm treatment in chronic wounds is debridement, and repeated debridement is often a critical principle of wound bed preparation [10,120]. Many methods of debridement exist, including sharp/surgical, larval, ultrasonic, hydrosurgical, autolytic/hydrogel, wet-to-dry, and enzymatic debridement [121,122]. Surgical/sharp wound debridement is recommended in human consensus guidelines, as limited evidence exist for other methods, e.g., ultrasonic and enzymatic debridement [10]. Similarly, in equine wound research, no specific guidelines exist on debridement, so to follow the human guidelines seems valid. An equine ex vivo study showed better effectiveness of hydrosurgical debridement compared to normal sharp debridement [123], thus hydrosurgery might be a useful subcategory of surgical debridement. No matter the method, infected tissue should be debrided whenever possible, and debridement should always be combined with other anti-biofilm strategies, as not all of the biofilms and bacteria can be removed [10,27]. Following debridement, a time-dependent therapeutic window opens (approximately 24 h), in which the biofilms re-establish themselves and mature. During this period, the biofilms will be easier to treat, and re-growth can potentially be repressed [119]. In our clinic, the surface of granulating wounds are often debrided at bandage change, in order to reduce bioburden and stimulate healing.

Topical treatment of biofilm wound infections after debridement is guided by the best available evidence (often “only” in vitro testing) and personal preferences, as no large prospective randomised clinical control trial exists [10,85]. A recent systematic review concluded that there is no evidence to recommend any topical therapy over another for treating biofilm in chronic wounds in humans [124]. Further, the British Equine Veterinary Association (BEVA) recently (2021) published a review article on primary care clinical guidelines for equine wound management [125], however no guidance on topical treatment of equine wounds is provided as the evidence is lacking. No studies have investigated the effect of any treatment on biofilms in equine wounds. It is beyond the scope of this review to present all potential topical treatments, but substances such as chlorhexidine, cadexomer iodine, silver (sulfadiazine or nanocrystalline), polyhexamethylene biguanide (PHMB, also called polyhexanide), high-osmolarity solutions (e.g., hypertonic saline, sugardine), honey, sodium hypochlorite, and acetic acid are among a long list of used potential anti-biofilm treatments [85,110,126]. More experimental topical therapies include quorum sensing inhibitors, bacteriophages, plant extracts, probiotic approaches, matrix breaking enzymes, immunomodulation therapy, etc. [31,111]. Negative pressure wound therapy with instillation of antimicrobial solutions is yet another treatment option [127,128]. Recently, a comprehensive overview was provided over the use of topical treatment in equine wound infections [110].

Despite intense research in animal models, the optimal anti-biofilm treatment has not yet been identified. In our clinic, we treat presumed biofilm infected wounds by repeated superficial debridements (sharp dissection with scalpel) followed by topical antimicrobials (Figure 2). The antimicrobials that are most effective are, in our experience, PHMB, nanocrystalline silver, and antibiotics. These may be applied as an integral part of commercially available dressings or ointments, or dressings containing the antimicrobial substance may be custom-made by soaking gauze or other dressing materials in the desired solution before applying them to the wound surface. Bandages are normally changed every 2 to 5 days based on exudate amounts and healing stage. Applying a hypertonic solution (hypertonic saline, sugar, or honey) daily for one to a few days may not only help reduce the bioburden, but also help remove necrotic areas and tenacious exudates. Wounds infected with *P. aeruginosa* present a particular challenge due to the inherently high resistance to antimicrobials in this organism, and its strong biofilm forming abilities [129]. *P. aeruginosa* infected wounds may benefit particularly from topical application of acetic acid [126], hypochlorous acid [130], or relevant antibiotics (e.g., aminoglycosides or fluoroquinoles). Bacteriological culture and assessment of antimicrobial resistance pattern is highly recommended, where *P. aeruginosa* infection is expected, as extensive resistance has been reported also in veterinary medicine [131]. For clinical cases, we have used daily applications of 2% acetic acid (gauze soaked in the solution and applied for 20 min. or longer) with no adverse effects (Figure 2).

## 6. Discussion

Bacteria reside as biofilms in wounds, also equine wounds. The role biofilm play in wound healing seems to be depended on other factors than just the bacteria, mainly the wound bed and its microenvironment, including inflammatory capacity, oxygen conditions, and healing pattern [10,18,84]. The competent healing of equine body wounds seems to be unaffected by the presence of biofilms, whereas the inferior healing of limb wounds is further protracted by the presence of biofilm [18]. These observations are based on experimental wound studies in horses, whereas only few studies have detected biofilm in equine accidental wounds. The true prevalence of biofilm in equine naturally occurring limb wounds is unknown, as optimal detection methods have not been used so far and few studies exist. From our experimental study, the prevalence of biofilm infection seems to be around 100% in equine limb wounds [17]. Further studies, employing the gold standard methods of biofilm detection, are essential to know the true prevalence and the consequences of biofilms in equine limb wounds.

Due to these findings of biofilms in equine limb wounds and the similarity to human chronic wounds, biofilms should be suspected in equine limb wound healing with delayed healing despite appropriate treatment. Debridement is the most effective therapy against biofilm in wounds, and fortunately, debridement is a well-established method in equine wound care, whereby exuberant granulation tissue can be debulked and the surface of infected/inflamed granulation tissue excised. This procedure will in most cases remove the majority of biofilm in the wound bed and thereby enhance healing. As mentioned above, no gold standard for topical treatment of biofilms in human chronic wounds exists to potentially guide equine topical treatment, however periodic debridement and topically applied antibacterial/anti-biofilm products seems to be the way forward.

## 7. Conclusions

Biofilms occur in equine limb wounds healing by second intention and are most likely a contributing factor to the delayed healing seen in limb wounds compared to body wounds. Unfortunately, no diagnostic tool for detection of biofilm in equine patient wounds exists yet. Until such tools are developed, biofilms should be suspected in wounds with delayed healing despite correct and appropriate treatment [10,110,132]. Even though biofilms cannot be directly diagnosed in routine samples currently, it is still relevant to know something about the bacterial burden (species present) of the wound and development over time to guide treatment. Biopsies are considered the gold standard for microbial diagnostics in wounds [100,102] and should be used for sequencing, PCR, and/or culture to get an overview of the different bacteria present in a given wound. If it is not possible to obtain a biopsy, swabs can in many cases also reveal relevant information when performed and handled correctly [12,101,133]. Treatment should be focused on repeated debridement in combination with topical application of antimicrobial therapies. Debridement serves to physically remove biofilms and other bacteria, thereby creating a healthier wound bed. The antimicrobial therapy serves to inhibit reestablishment of biofilm after debridement.

## Figures and Tables

**Figure 1 animals-11-02825-f001:**
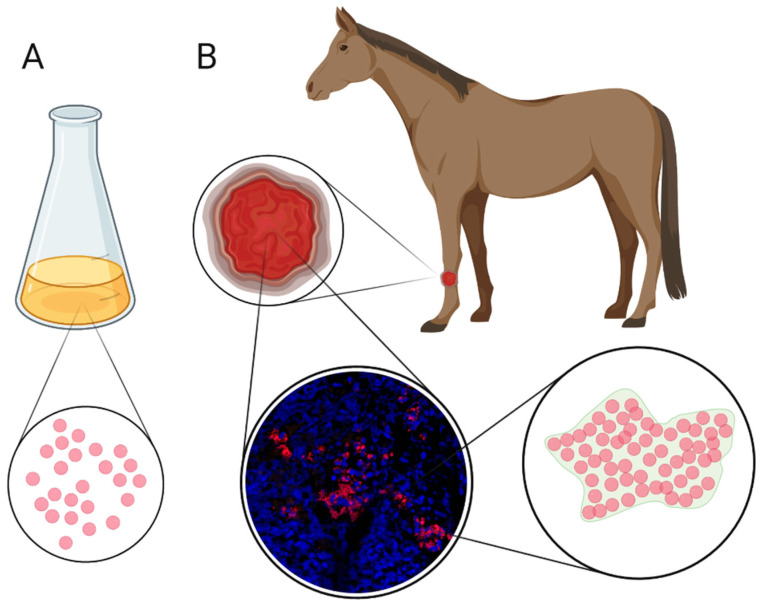
(**A**) Normally bacteria growing in shaken cultures in the lab are planktonic (single cells); (**B**) Bacteria in most situations in nature, including wounds, are present as biofilms. Biofilms are aggregates of bacteria, and within these biofilms, the bacteria are protected from immune defences and antimicrobial products. The microimage shows bacterial biofilm in an equine limb wound biopsy detected with peptide nucleic acid (PNA) fluorescence in situ hybridization (FISH) and subsequent confocal laser scanning microscopy (CLSM). Bacterial ribosomal RNA is displayed with a red fluorophore and eukaryotic cell nuclei (DNA) are blue due to DAPI stain (4′,6′-diamidino-2-phenylindole). Many eukaryotic cell nuclei (seen as blue spots) surround the biofilms and leaked DNA (blue stain not shaped as nuclei) from the cells can be seen, this leaked DNA is most likely from neutrophils dying in the combat against the biofilms. Figure created with BioRender.com (accessed on 7 July 2021).

**Figure 2 animals-11-02825-f002:**
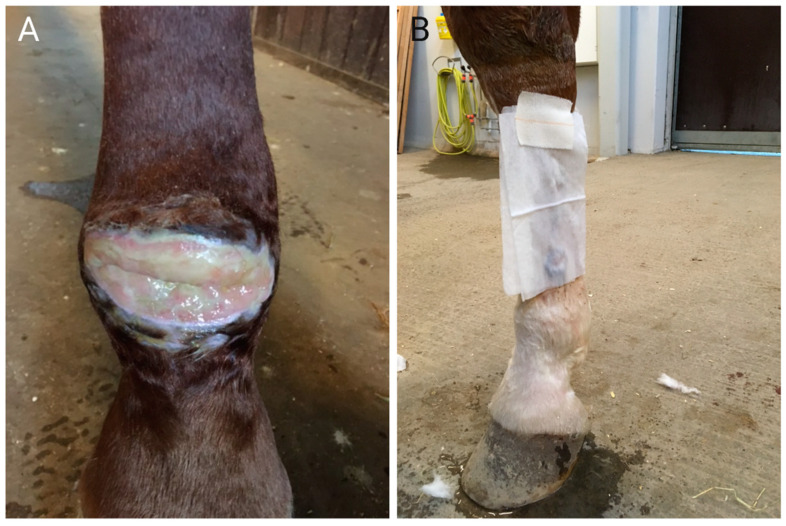
(**A**) Wound with exudate and presumed infection with biofilm. This wound will need reduction of bioburden to heal. This may be achieved by surgical debridement of the wound surface followed by topical application of antimicrobials (e.g., dressings containing cadexomer iodine, nanocrystalline silver, silver sulfadiazine, polyhexamethylene biguanide, or acetic acid). The treatment/dressing is reapplied at each bandage change, normally performed every 2 to 5 days. Reduction of bioburden and removal of exudates may also occur through application of dressings containing hypertonic saline; these are applied daily for one or a few days, where after the wound is re-evaluated. When treating wounds with crevices in the wound surface, it is important to ensure debridement of the crevice and to pack dressings into the crevice to ensure that the entire wound surface is in contact with the antimicrobial substance; (**B**) Application of acetic acid to a limb wound. Acetic acid soaked gaze is secured over the wound (cling wrap is applied on top to further secure), the gaze can be kept soaked by multiple applications of acetic acid (1–2%) over a 20–30-min treatment period.

## Data Availability

Not applicable.

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
