# Peer review of "Biofilm and Equine Limb Wounds"

_animals, 2021, doi:10.3390/ani11102825_

Round 1

Reviewer 1 Report

Thank you for an interesting and well-organized manuscript.  The figures are of good quality.  Length of the manuscript is appropriate.  Well referenced.  I recommend the manuscript for publication.

I particularly appreciated your attention to clarifying the difference between antimicrobial tolerance and antibiotic resistance.  The points were important and clear.

Moderate English editing is needed to ensure subject-verb agreement, appropriate tense, ease of reading, and to address a few spelling errors.  

Author Response

Dear reviewer

Thank you for evaluating the manuscript and the kind words. English editing has been performed, please see the updated manuscript and replies to comments from reviewers.

Reviewer 2 Report

Interesting paper that deals with an extremely current topics. The article is well structured, however requires some revision and some addition.

In particular:

  • Line 54-58 reword, the sentence appears confusing
  • Line 61-64 add references
  • Line 163-166 They talk about the pig suddenly explain well the link between species
  • Line 290 "Treatment of biofilm in wounds"..... see attached file

Author Response

Dear reviewer

Thanks for taking your time to evaluate our papar, we highly appreciate this.

Replies to comments:

  • Interesting paper that deals with an extremely current topics. The article is well structured, however requires some revision and some addition.

Reply: Thanks for your comment! As seen below in the replies to your comment, and in the updated manuscript and in the response to reviewer 3, many changes/improvement have been made to this document.

In particular:

  • Line 54-58 reword, the sentence appears confusing

Reply: Thanks for pointing this out, we have amended the section accordingly: “Bacteria can live either as single cells, also known as planktonic cells, or as biofilm (Fig 1). Most of our knowledge of bacteria originates from classic culture plates and shaken cultures where the bacteria live as planktonic cells. Nevertheless, biofilm formation is an ancient aspect of bacterial life and one of the reasons bacteria can be found throughout nature, even in extremely hostile and depleted environments. Thus, biofilm is the default mode of bacterial life. Unfortunately, biofilm is a more complicated lifestyle to study than planktonic cells [22] and the majority of our knowledge comes from in vitro studies of biofilms.”

(Original text is shown in italics) Bacteria can live either as single cells, also known as planktonic cells, or as biofilm (Figure 1). Most of our knowledge of bacteria originates from classic culture plates and shaken cultures where the bacteria live as planktonic cells. Nevertheless, biofilm is the default mode of bacterial life, and as this is a more complicated lifestyle to study [22] the majority of our knowledge comes from in vitro studies of biofilms.

 Line 61-64 add references

Reply: The reference is given in the sentence before: “A recent international consensus statement [26] defined biofilm as:…”, so it is ref 26. (Haesler, E.; Swanson, T.; Ousey, K.; Carville, K. Clinical indicators of wound infection and biofilm: reaching international consensus. J Wound Care 2019, 28, s4-s12, doi:10.12968/jowc.2019.28.Sup3b.S4.)

  • Line 163-166 They talk about the pig suddenly explain well the link between species’

Reply: We have tried to ease the transition from rodents/rabbits to pigs in the text by adding: “Another more relevant, however less used, model animal in wound research is the pig, as porcine wound healing resembles human wound healing to a larger extent than rodents and rabbits because pigs are more tight-skinned and heal predominantly by epithelialisation [79-81]…”

(original text in italics)Porcine wound healing resembles human wound healing to a larger extent than rodents and rabbits because pigs are more tight-skinned and heal predominantly by epithelialisation [79-81]; accordingly porcine wound models are considered superior to other animal models due to their translational value

 Line 290 "Treatment of biofilm in wounds"+ attached file:

  • Being a review you don't have to limit yourself to your wound treatment but talk extensively about all types of existing debridement eg. biological, autolytic etc. etc. this paragraph must be rewritten entirely.

Reply: Thanks for your input. The debridement part has been updated to include the guidelines for human chronic wounds with biofilm (Surgical/sharp wound debridement is recommended in human consensus guidelines, as limited evidence exist for other methods, e.g. ultrasonic and enzymatic debridement), and relevant methods has been listed: “The cornerstone of biofilm treatment in chronic wounds is debridement, and repeated debridement is often a critical principle of wound bed preparation [10,120]. Many methods of debridement exist including sharp/surgical, larval, ultrasonic, hydrosurgical, autolytic/hydrogel, wet-to-dry, and enzymatic debridement [121,122]. Surgical/sharp wound debridement is recommended in human consensus guidelines, as limited evidence exist for other methods, e.g. ultrasonic and enzymatic debridement [10]. Similarly, in equine wound research, no specific guidelines exist on debridement, so to follow the human guidelines seems valid. An equine ex vivo debridement study showed better effectiveness of hydrosurgical debridement compared to normal sharp debridement [123], so hydrosurgery might be a valid subcategory of surgical debridement. No matter method, infected tissue should be debrided whenever possible, and debridement should always be combined with other anti-biofilm strategies, as not all of the biofilms and bacteria can be removed [10,27]. Following debridement, a time-dependent therapeutic window opens (approximately 24 hours), in which the biofilms re-establish themselves and mature. …”. Further other changes has been made to this section as requested by reviewer 3.

Reviewer 3 Report

This review deals with an important aspect of orthopaedic infections, the presence of biofilm formation in chronic and non-healing wounds. While I believe that this is a topic which is certainly suitable and deserving for a review article, there are several concerns that need to be addressed, by the authors, before I recommend its publication in Animals.

General comments

This reviewer has a number of concerns with the manuscript in its present format, and offers the following comments for consideration.

  1. While some sections of the manuscript are well written and have very little grammar and spelling mistakes, other sections contain sentences that warrant proper english editing and perhaps a professional english editing service might be helpful prior to resubmission.

  1. The review lacks conciseness. There are several sections which repeat the same statements (e.g. introduction and biofilm and equine limbs). I believe the review would greatly benefit from shortening the initial sections 1-4 and expanding detail and description of current treatments and on research regarding new treatment approaches (based on human research).

  1. As state already above. There is a lack of discussion on the latest approaches for management of biofilms in chronic wounds. I would also like to see some discussion on alternative debridement strategies aside from surgical, sharp dissection. The authors state that it is beyond the scope of this review to present all potential topical treatment options (line 314-315) but I would actually expect at least a review of the most commonly used once, and not only list them but add information on application, dosages, frequency and when to change to a different topical dressing or strategy. I understand that there is limited evidence in equids at this point but since the authors state this limitation at the beginning of this section they could/should extrapolate (as done in general in this section) from other species. Instead of citing a book chapter as a recent comprehensive review, I would expect this to be present here to a certain extent (line 320).

  1. Quality of the images is insufficient. Particularly the microimage of the biofilm is insufficient to identify the depict features.

Specific comments:

Simple Summary:

Please try to shorten the current information and make them more concise (this could be written in 2-3sentences) and add more information on later sections of your review and or the goal/purpose of your review.

Abstract:

Line 21: biofilms are the default mode....

Line 23: exchange often by commonly

Line 29: and several times throughout the review the authors refer to infections with biofilm forming bacteria as biofilm infection. This is mostly for consideration and a general comment: Is it not still the bacteria that cause the infection and therefore it would be more appropriate to refer to these infections as infection with biofilm producing bacteria or presence of a biofilm in infected wounds?

Lines 29-31: This sentence reads a little awkward and contains twice the word treatment. Please rephrase/change.

Line 32: Key words should usually not include the same words as the title so that search engines can use them. I would suggest to change the once you already got in your title to e.g. horse instead of equine,…perhaps delayed wound healing…

Main manuscript:

Line 37 and line 42: see comment above regarding biofilm infection.

Line 47: affect instead of affected. And wounds healing of distal limb wounds but not body wounds.

Line 48: Sentence reads awkward, please rephrase. Also, perhaps refer to healing of chronic wounds as delayed wound healing?

Lines 51-51: perhaps etiology and prevalence?

Lines 56-58: I believe this is meant to be two sentences separated after the reference 22. Otherwise please rephrase so that it makes sense. It would also be good instead of just stating that it is the default mode of life several times in your review to explain this statement in one or two sentences.

Line 67: I believe you mean the bacteria and not the biofilm as the matrix is part of the biofilm. The matrix provides the bacteria within the biofilm with protection…

Lines 84-94: Figure 1: In general a nice image, unfortunately, particularly the last part of your legends description about the neutrophils is impossible to identify based on the provided image. It would be good to add a better microscopic image of a biofilm and it could be a separate image/figure.

Lines 113-115: This is not proper scientific English, please rephrase.

Lines 131-134: Please consider rephrasing it again does not sound very scientifically sound.

Lines 135-136: Consider changing the name of this paragraph to better reflect the content.

Otherwise the initial two thirds of this section are interesting and well written.

Line 168: change: exudates to exudate

Lines 171-174: This is a long sentence and contains twice “however”, you could consider changing the second however to but. I would also consider to tie this sentence to earlier statements about biofilm formation

Line 174-176: When you make a statement like this regarding the differences it is always good if you have previously explained what the human patient population looks like. I do not think you can expect the reader to know that these wounds are most common in old and chronically ill people with multiple co-morbidities.

Lines 176-179: Please revise this section, as it reads awkward and not very scientifically sound.

Line 180: delays

Line 185: This section has considerable overlap with other part of the manuscript, e.g. introduction. Please try and make it more concise and focus on the main aspects and rather include a bit more detail on certain aspects than repeat stating facts that you already mentioned (e.g. lines 198-200).

Line 188: You may need to explain what you mean with suboptimal detection methods, as a reader may not know these studies by heart.

Lines 190-191: Sentence reads awkward please revise.

Line 198: What is the gold standard? Please explain here as it is not clear to a reader.

Line 201-206: A larger degree of contamination on distal limb wounds has been proposed in the past as well.

Lines 212-215: Please revise sentence, as it reads currently awkward.

Line 216-220: This section again does contain a lot of overlapping information that has been state previously and writing style requires revision.

Line 224-225: Consider deleting the last sentence of this introduction to this section.

Line 237: as

Line 242-265: Good information but try and organise it in a more logical manner as it is currently. E.g. 1. Why is the diagnosis generally challenging, 2. What are the specific difficulties with culture based methods, 3. What are different sampling methods and their advantages and disadvantages and technical limitations. Or whatever you see fit, but currently it is still not very well organised. The writing style should also be reconsidered here.

Line 266-271: If you want to talk about using sequence based techniques I think you would need to expand this as depends highly on the technique used whether or not you will gain valuable information. For example general 16srRNA sequencing may not give you sufficient sequencing depth to identify bacteria on species level. But there is a relatively recent study that using metagenomic shot gun sequencing to assess bacteria at species level and also some of the resistant genes present (Kalan et al., 2019). I believe it would be good to expand this section as it is an evolving field and may prove very valuable.

Line 281-282: and instead of but

Lines 281-289: Is there no more information on modalities that are currently explored? I think this section would benefit from additional information.

Line 306: Is this “time-dependent therapeutic window” 24h? please state clearly as it is useful for a clinician dealing with this type of wounds.

Line 302-310: It would be good to add some detail on wound debridement. Is there other option aside from surgical, sharp wound debridement (I believe there is) and if so what are their advantages and disadvantages and how to apply them.

Lines 311-352: As I already mentioned in the general comments, I believe this section need more detail and it is definitely within the scope to talk about the most commonly used topical treatment options in more detail. I would also add the currently explored treatment options in human medicine (there is studies on use of e.g. N-acetyl-cystein, alpha-topopherol,… bacteriophage therapy,… and also use of negative pressure wound therapy just to name a few) and I do not think it is appropriate to cite another review/book as a comprehensive review of topical treatments of wound infections in horses as this is specific for wounds where biofilm formation is suspected. It could be considered to add a table with the most common topical treatments and details (e.g. concentration, frequency of change of dressings, duration of treatment and when to discontinue treatment,…).

Line 329: awkward what of stating the frequency, also what does that now mean exactly? –Every day or every other day. Please be more precise about your description of treatment and management. This sentence is also in the legend of figure 2 and here applies the same.

Line 330-332: Please revise sentence for proper English grammar.

Lines 332-339: This is interesting but as P. aeruginosa is not the only pathogen it would be good to add some more common once and their treatment regiments.

Figure 2: I know it is sometimes difficult to get good quality images but it would be nice to use images with a cleaner background for this type of publication.

Line 361: naturally occurring wounds

Lines 362-365: Sentence contains twice experimental/experimentally, it would be good to find a different way of stating this.

Line 362-370: This section has again not a very good scientific writing style and I would consider revising. It also includes again information that have been mentioned several times before in this manuscript. Please try and make an effort to eliminate repetition of information in your manuscript in general.

Author Response

Dear Reviewer

We thank you for your great input and throrougness in evaluating our paper. We have adressed all comments in the attached file. For your ease, the original text is shown in italics (when needed).

We find the manuscript highly improved.

Round 2

Reviewer 1 Report

Thank for the improvements made to the English grammar.  I found the manuscript improved in readability and greatly appreciate your efforts.  I support the publication of the manuscript and its contents and found only very minor grammar issues that can be easily addressed by the editorial staff and do not require re-review by me.

Congratulations and good luck to the PhD student who did much of the work.

Lay summary - the final two sentences are slightly awkward in English.  I suggest the following:

Line 15:  delete the word "further".  

Lines 16-18:  reword to something like:  ". . . physically removing biofilm and unhealthy tissue from the wound with surgery and immediately applying antimicrobial compounds to kill any biofilm or bacteria not removed by surgery.

Subject-verb disagreement should be corrected on line 270:

"swabs poses" - please remove the final letter s from poses

Another awkward spot on line 290 - "methods do also not provide" - I suggest deleting the word "also"

Possessive is not needed on lines 351-352:  "the British Equine Veterinary Association's (BEVA) recently (2021) published" - please remove the apostrophe and s on the end of Association

Author Response

Dear Reviewer

Thanks for the nice comments and the throrough read-through to catch small grammar issues.

We have corrected as suggested by you.

Reviewer 3 Report

Review: animals-1312554: Biofilm and equine limb wounds (review article)

I believe the manuscript has been markedly improved since I read it last. There are still a few awkward sentences that would require English grammar correction but the style is in general appropriate for publication. The authors also added some previously missing information. The aim of the review as stated at the end of the introduction is fulfilled by the content of the manuscript. But as there is still so limited information on this topic in equine wounds, I would really like some more information and discussion on further (or perhaps future) option on improving detection and treatment of biofilms. It just seems very basic to say, that biofilm formation should be suspected and debridement and antimicrobial therapy should be used, as this is already common practice in chronic distal limb wounds in horses.

Ultimately, I believe that the authors have done a good job in summarising the current knowledge and the review may be of value for equine practitioners.

Author Response

Dear Reviewer

We thank you for the kind words and previous inputs, which have improved the manuscript.